# Transcriptome-wide identification of 5-methylcytosine by deaminase and reader protein-assisted sequencing

**Jiale Zhou[1†], Ding Zhao[1,2†], Jinze Li[1,2†], Deqiang Kong[1], Xiangrui Li[1], Renquan Zhang[1], Yuru Liang[1], Xun Gao[1], Yuqiang Qian[1], Di Wang[1], Jiahui Chen[1], Liangxue Lai[1]\*, Yang Han[1]\*, Zhanjun Li[1,2,3]\***

[1]Jilin Provincial Key Laboratory of Animal Embryo Engineering, State Key Laboratory for Diagnosis and Treatment of Severe Zoonotic Infectious Diseases, Key Laboratory for Zoonosis Research of the Ministry of Education, Jilin University, Changchun, China; [2]Laboratory of Organ Regeneration and Transplantation of The Ministry of Education, State Key Laboratory for Diagnosis and Treatment of Severe Zoonotic Infectious Diseases, First Hospital of Jilin University, Changchun, China; [3]Sanya Institute of Swine Resource, Hainan Provincial Research Center of Laboratory Animals, Sanya, China

**\*For correspondence:**
lai_liangxue@gibh.ac.cn (LL);
hanyang8584@jlu.edu.cn (YH);
lizj_1998@jlu.edu.cn (ZL)

[†]These authors contributed equally to this work

**Competing interest:** The authors declare that no competing interests exist.

## eLife Assessment

This potentially **useful** study introduces an orthogonal approach for detecting RNA modification, without chemical modification of RNA, which often results in RNA degradation and therefore loss of information. Compared to previous versions, the most recent one is improved and sufficiently aligned with the standards of the field to merit consideration by the research community, making the evidence **solid** according to said standards. Nevertheless, uncertainty regarding false positive and false negative rates remains, as it does for some of the alternative approaches. With more rigorous validation, the approach might be of particular interest for sites in RNA molecules where modifications are rare.

**Abstract** 5-Methylcytosine (m5C) is one of the posttranscriptional modifications in mRNA and is involved in the pathogenesis of various diseases. However, the capacity of existing assays for accurately and comprehensively transcriptome-wide m5C mapping still needs improvement. Here, we develop a detection method named DRAM (deaminase and reader protein assisted RNA methylation analysis), in which deaminases (APOBEC1 and TadA-8e) are fused with m5C reader proteins (ALYREF and YBX1) to identify the m5C sites through deamination events neighboring the methylation sites. This antibody-free and bisulfite-free approach provides transcriptome-wide editing regions which are highly overlapped with the publicly available bisulfite-sequencing (BS-seq) datasets and allows for a more stable and comprehensive identification of the m5C loci. In addition, DRAM system even supports ultralow input RNA (10 ng). We anticipate that the DRAM system could pave the way for uncovering further biological functions of m5C modifications.

## Introduction

Epigenetics refers to stable inheritance without changing the basic sequence of DNA, involving various forms such as DNA methylation, histone modification, and RNA modification. In recent years, RNA

sequencing technology has boosted research on RNA epigenetics. More than 170 RNA modifications have been identified, mainly including m$^6$A, m$^5$C, m$^1$A, m$^7$G, and others (***Wiener and Schwartz, 2021***; ***Li and Mason, 2014***). Notably, RNA m$^5$C methylation represents a crucial posttranscriptional modification observed across different RNA types, such as tRNA, mRNA, rRNA, vault RNA, microRNA, long noncoding RNA, and enhancer RNA (***Van Haute et al., 2019***; ***Blaze et al., 2021***; ***Yang et al., 2017***; ***Sharma et al., 2013***; ***Hussain et al., 2013***; ***Aguilo et al., 2016***). Numerous studies have revealed multiple molecular functions of m$^5$C in numerous key stages of RNA metabolism, such as mRNA stability, translation, and nuclear export (***Yang et al., 2017***; ***Yang et al., 2019***; ***García-Vílchez et al., 2019***; ***Trixl and Lusser, 2019***; ***Boo and Kim, 2020***; ***Courtney et al., 2019***). The dynamic alterations of m$^5$C play integral roles in many physiological and pathological processes, such as early embryonic development (***Liu et al., 2022***), neurodevelopmental disorder (***Chen et al., 2019b***; ***Flores et al., 2017***), and multifarious tumorigenesis and migration (***Chen et al., 2022***; ***Chen et al., 2019a***; ***Yang et al., 2021***; ***Li et al., 2023***). Moreover, this modification significantly contributes to the regulation of gene expression (***Yang et al., 2017***; ***Yang et al., 2019***; ***García-Vílchez et al., 2019***; ***Trixl and Lusser, 2019***; ***Boo and Kim, 2020***; ***Courtney et al., 2019***; ***Chen et al., 2022***). Therefore, the detection of m$^5$C sites appears to be essential for understanding their underlying effects on cellular function and disease states.

With the recent advances in sequencing techniques, several high-throughput assays have been developed for qualitative or quantitative analysis of m$^5$C. To date, bisulfite-sequencing (BS-seq) has been proven to be the gold standard method for RNA m$^5$C methylation analysis (***Yang et al., 2017***; ***Huang et al., 2019***; ***Trixl et al., 2019***). This approach chemically deaminates unmethylated cytosine to uracil, while keeping methylated cytosine unchanged. The m$^5$C methylation sites can be identified by subsequent library construction and sequencing. However, bisulfite treatment of BS-seq is extremely detrimental to RNA, thus resulting in unstable detection of m$^5$C in low abundance RNA or highly structured RNA, which directly affects the confidence of results (***Schaefer et al., 2009***; ***Amort and Lusser, 2017***). Another major type of global m$^5$C analysis depends on antibody-assisted immunoprecipitation of m$^5$C-methylated RNAs, such as m$^5$C-RIP-seq (***Cui et al., 2017***; ***Edelheit et al., 2013***; ***Saplaoura et al., 2020***), AZA-IP-seq (***Khoddami and Cairns, 2013***), or miCLIP-seq (***Hussain et al., 2013***). These methods are unable to recognize methylation on mRNAs with low abundance and secondary structure. Moreover, these methods are highly dependent on antibody specificity, which usually leads to unspecific binding of RNA and a low amount of m$^5$C-modified regions. Moreover, TAWO-seq, originally developed for the identification of hm$^5$C, is also capable of m$^5$C analysis, but it highly depends on the oxidation efficiency of perovskite, which usually causes false positives and unstable conversion (***Yuan et al., 2019***; ***Hayashi et al., 2016***). Furthermore, the emerging third-generation sequencing, such as Nanopore-seq, can directly map m$^5$C by tracking the characteristic changes of bases, but it still faces challenges of a high error rate (***Harel et al., 2019***; ***Chang et al., 2021***; ***Wan et al., 2022***). These together largely hamper its wide application on transcriptome profiling of m$^5$C (***Supplementary file 1***). Hence, there is an urgent need for a simple, efficient, sensitive, and antibody-independent method for global m$^5$C detection.

The RNA-binding protein ALYREF is the initially recognized nuclear m$^5$C reader that binds directly to m$^5$C sites in mRNA and plays key roles in promoting mRNA nuclear export or tumor progression (***Yang et al., 2017***). Another well-known m$^5$C reader, YBX1, can also recognize m$^5$C-modified mRNA through its cold-shock domain and participates in a variety of RNA-dependent events such as mRNA packaging, mRNA stabilization, and translational regulation (***Yang et al., 2019***; ***Chen et al., 2019a***). RNA affinity chromatography and mass spectrometry analyses using biotin-labeled oligonucleotides with or without m$^5$C were performed in previous reports, which indicated that ALYREF and YBX1 had a more prominent binding ability to m$^5$C-modified oligonucleotides (***Yang et al., 2017***; ***Chen et al., 2019a***). YBX1 can preferentially recognize mRNAs with m$^5$C modifications via key amino acids W65-N70 (WFNVRN) (***Chen et al., 2019a***), while K171 is essential for the specific binding of ALYREF to m$^5$C sites (***Yang et al., 2017***). Previous studies have shown that mutations in key amino acids responsible for recognizing m$^5$C binding in ALYREF and YBX1 lead to a significant reduction in their binding levels to m$^5$C-containing oligonucleotides (***Yang et al., 2017***; ***Chen et al., 2019a***). Nucleic acid deaminases, primarily categorized as cytosine deaminases and adenine deaminases, are zinc-dependent enzymes which facilitate the deamination of cytosine or adenine within DNA or RNA substrates (***Budzko et al., 2023***). APOBEC1, an evolutionarily conserved family member of APOBEC proteins, can specifically

catalyze the deamination of cytosine in single-stranded RNA (ssRNA) or DNA (ssDNA) to uracil (*Rosenberg et al., 2011*; *Qian et al., 1998*; *Siriwardena et al., 2016*). TadA8e is an adenine deaminase optimized through reengineering of TadA and it induces conversion of adenine to inosine (eventually read as guanine by transcriptases) in ssRNA or ssDNA (*Richter et al., 2020*; *Gaudelli et al., 2020*). APOBEC1 and TadA8e, with their prominent deamination efficiency, have been employed for the development of precise and efficient base editors such as CBE and ABE8e, which find widespread application in studies related to genome editing (*Siriwardena et al., 2016*; *Richter et al., 2020*).

Here, we aim to establish a deaminase and m5C reader-assisted RNA methylation sequencing approach (DRAM-seq), which identifies the m5C sites through reader-mediated recognitions and deaminase-mediated point mutations neighboring the m5C methylation sites. This bisulfite-free and antibody-free method is anticipated to provide more comprehensive and cost-effective transcriptome-wide detection of m5C methylation, which may better assist on exploring its further regulatory mechanisms.

## Results

### Development of DRAM system for m5C detection

Our sequencing platform is inspired by the concept of the m6A DART-seq assay, in which C near the m6A site is converted into U without affecting sequences near non-m6A sites (*Meyer, 2019*). Therefore, we hypothesized that, by utilizing the targeted binding of m5C readers, deaminase can be recruited to achieve deamination of cytosine or adenine in the vicinity of the m5C sites on ssRNA, thereby facilitating the detection of the m5C site. This approach was named DRAM (deaminase and m5C reader-assisted RNA methylation sequencing). As RNA-binding proteins, ALYREF and YBX1 also could bind to RNAs without m5C modification (*Yang et al., 2017*; *Chen et al., 2019a*). To exclude the false-positive detection of DRAM due to the non-m5C-specific binding of ALYREF and YBX1, knockout

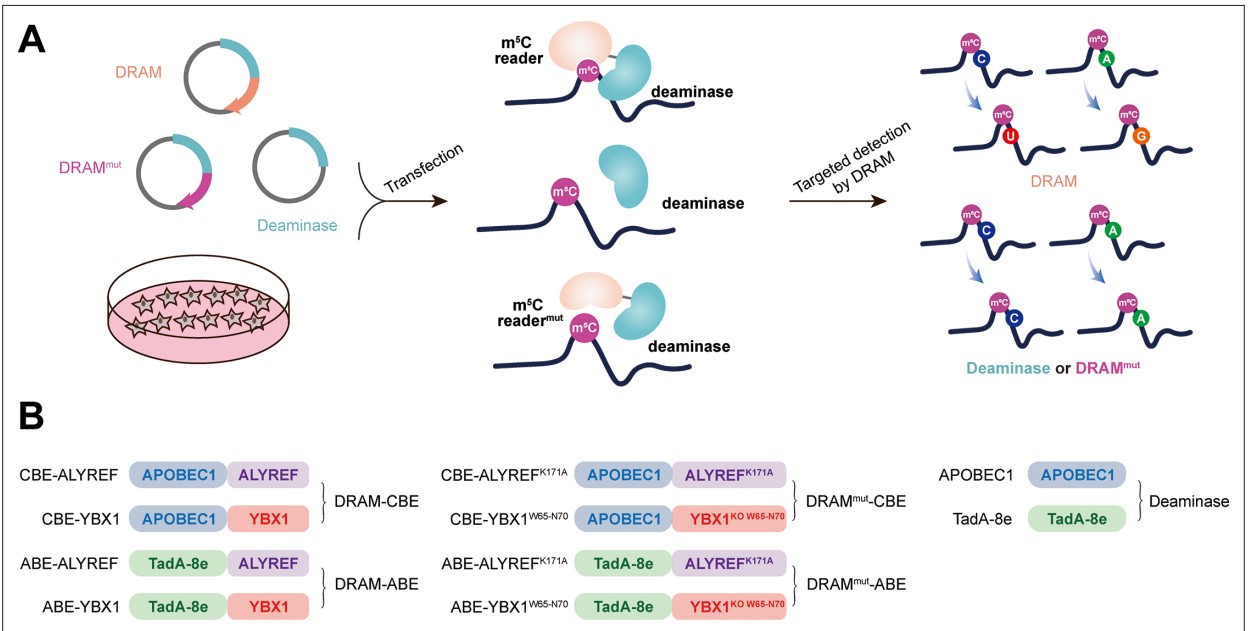

**Figure 1.** Development of DRAM system for 5-methylcytosine (m5C) detection. (**A**) Schematic diagram of the DRAM assay. DRAM, DRAM^mut, and Deaminase system were transfected into HEK293T cells separately. After DRAM transfection, the deaminase was directed by m5C reader to the vicinity of the m5C site and induce C-to-U/A-to-G mutations, whereas transfection of the DRAM^mut or Deaminase system failed to effectively induce similar mutations due to the absence of the m5C-recognition-binding domain. (**B**) The overall design of DRAM, DRAM^mut, and Deaminase system.

The online version of this article includes the following source data and figure supplement(s) for figure 1:

**Figure supplement 1.** The three-dimensional (3D) structures of ALYREF^K171A, YBX1^KO W65-N70, ALYREF, and YBX1.

**Figure supplement 1—source data 1.** PDF file containing original western blots for *Figure 1—figure supplement 1E-H*, indicating the relevant bands and treatments.

**Figure supplement 1—source data 2.** Original western blots for *Figure 1—figure supplement 1E-H*.

of W65-N70 (WFNVRN) amino acids in YBX1 and K171A mutation in ALYREF were introduced separately, resulting in the DRAM$^{mut}$ system (*Figure 1—figure supplement 1A–D*). Subsequently, we verified the affinity ability of YBX1 and ALYREF for m$^5$C-modified RNAs by RNA pull-down experiments. Consistent with previous reports (*Yang et al., 2017*; *Chen et al., 2019a*), those two m$^5$C readers preferentially bound RNAs containing m$^5$C modifications. Furthermore, mutating key amino acids involved in their interaction with m$^5$C significantly reduced their binding ability, indicating that ALYREF and YBX1 exhibit specificity for m$^5$C-methylated mRNAs (*Figure 1—figure supplement 1E–H*). To confirm the recognition of m$^5$C site by DRAM system, DRAM, DRAM$^{mut}$, and Deaminase system were transfected into the human HEK293T cells, respectively. Finally, we considered the presence of m$^5$C modification in the vicinity only if the deamination changes produced under DRAM induction were significantly different from those produced under DRAM$^{mut}$ or Deaminase induction (*Figure 1A*).

Previous studies have indicated that there is no uniform intrinsic signature motif sequence that can characterize all m$^5$C sites (*Yang et al., 2017*; *Edelheit et al., 2013*; *Sun et al., 2019*; *Zhang et al., 2020b*). To comprehensively detect the m$^5$C loci, the readers of m$^5$C (ALYREF and YBX1) were separately fused to the C-terminus of the deaminases (APOBEC1 and TadA-8e), namely DRAM-ABE and DRAM-CBE system (*Figure 1B*).

## DRAM detection system is assayed in an m$^5$C-dependent form

To confirm the recognition of m$^5$C site by DRAM system, DRAM, DRAM$^{mut}$, and Deaminase were transfected into the human HEK293T cells, respectively. To evaluate candidate DRAM constructs within a cellular environment, we performed fluorescence microscopy to analyze the expression of DRAM. The results showed that DRAM-ABE and DRAM-CBE were properly expressed in HEK293T cells (*Figure 2—figure supplement 1A and B*). In addition, flow cytometry that displayed ~60% of cells were GFP-positive (*Figure 2—figure supplement 1C*). Two previously reported m$^5$C sites in RPSA and AP5Z1 were selected for the analysis (*Yang et al., 2017*; *Huang et al., 2019*), and their methylation status was verified by BS-seq PCR. The deep sequencing results showed that the m$^5$C fraction of RPSA and SZRD1 was 75.5% and 27.25%, respectively (*Figure 2A and B*). Sanger sequencing following RT-PCR was then performed to determine the editing of neighboring m$^5$C sites by DRAM system in these two mRNAs. Notably, adenine close to the m$^5$C site in RPSA mRNA was mutated into guanine, resulting in an A-to-G editing rate of 14.7% by DRAM-ABE, whereas this was rarely observed with TadA-8e or DRAM$^{mut}$-ABE (*Figure 2C*). DRAM-CBE induced C-to-U editing in the vicinity of the m$^5$C site in AP5Z1 mRNA, with 13.6% C-to-U editing, while this effect was significantly reduced with APOBEC1 or DRAM$^{mut}$-CBE (*Figure 2D*). Subsequently, in order to investigate whether the DRAM system can detect other types of RNA, such as tRNA, 28S rRNA, or others, we performed PCR amplification of the flanking sequences of the m$^5$C sites 3782 and 4447 on 28S rRNA and several m$^5$C sites on tRNA, such as the m$^5$C48 and m$^5$C49 sites of tRNA$^{Val}$, the m$^5$C48 and m$^5$C49 sites of tRNA$^{Asp}$, and the m$^5$C48 site of tRNA$^{Lys}$. But Sanger sequencing showed that there was no valid A-to-G/C-to-U mutation detected, which is most likely due to the fact that ALYTEF and YBX1 are mainly responsible for the mRNA m$^5$C binding proteins, and thus the DRAM system is more suitable for the mRNA m$^5$C detection (*Figure 2—figure supplement 2*). Taken together, the fusion of m$^5$C reader and deaminase can effectively and selectively deaminate cytosine/adenine in the vicinity of the mRNA m$^5$C sites.

NSUN2 (*Zhang, 2021*) and NSUN6 (*Liu et al., 2021*), two family members of NOL1/NSUN protein, were both identified as m$^5$C methyltransferase of mRNA (*Schumann et al., 2020*). To verify that the detection of DRAM occurs in the presence of m$^5$C, we performed knockdown experiments of NSUN2 and NSUN6 in HEK293T cells by base deletion, resulting in frameshift mutations that led to reduced expression of NSUN2 and NSUN6. These cells were then transfected with DRAM. The knockout efficiency has been confirmed by western blotting (*Figure 2E and F*, *Figure 2—figure supplement 3A and B*). It has been previously demonstrated that m$^5$C methylation of AP5Z1 and RPSA is catalyzed by NSUN2 and NSUN6, respectively (*Huang et al., 2019*; *Fang et al., 2020*). In line with this, Sanger sequencing following RT-PCT showed a significant reduction in C-to-U or A-to-G mutations near the m$^5$C sites in methyltransferase-deficient cells compared with WT cells (*Figure 2G and H*). Overall, these findings suggest that the DRAM detection system is assayed in an m$^5$C-dependent form.

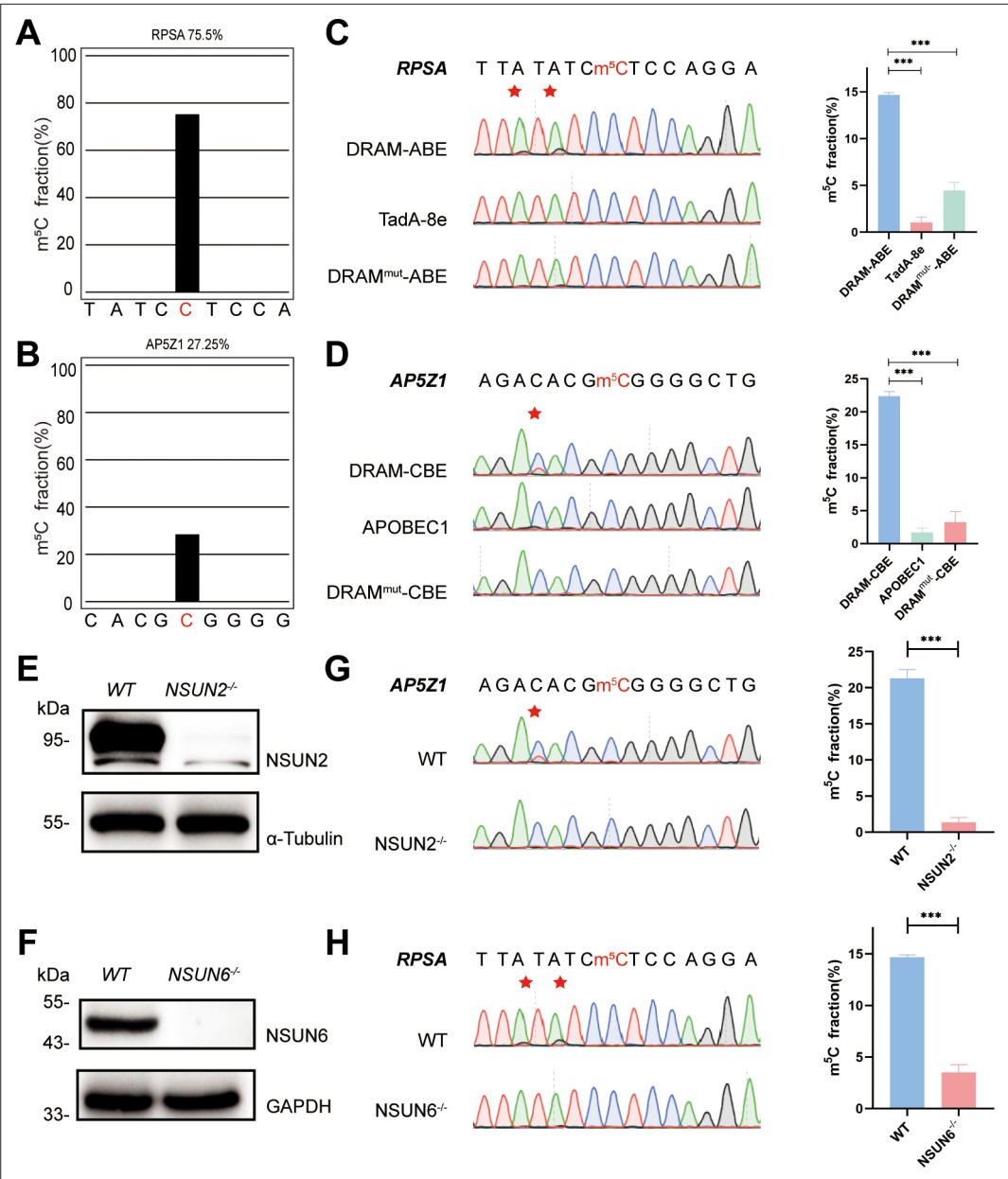

**Figure 2.** DRAM detection system was assayed in an 5-methylcytosine (m5C)-dependent form. (**A, B**) Two m5C sites from RPSA (**A**) and AP5Z1 (**B**) mRNA detected by deep sequencing of bisulfite-sequencing PCR in HEK293T cells. The m5C sites are highlighted by red color. The m5C fraction of RPSA and AP5Z1 were 75.5% and 27.25% (the number of reads is greater than 1000). (**C, D**) Sanger sequencing following RT-PCR verified two m5C sites from RPSA (**C**) and AP5Z1 (**D**) mRNAs in DRAM-transfected HEK293T cells, respectively. HEK293T cells only expressing DRAM^mut or Deaminase were served as negative controls. The left panel illustrates the location of DRAM-induced mutation sites, which is highlighted in red asterisk. The right panel shows the corresponding quantification of Sanger sequencing (n=3). (**E, F**) The knockout efficiency of NSUN2 (**E**) and NSUN6 (**F**) in HEK293T cell lines verified by western blotting. The protein level of α-Tubulin and GAPDH were served as loading controls, separately. (**G, H**) DRAM-induced mutations close to m5C sites in AP5Z1 (**G**) and RPSA (**H**) mRNAs after NSUN2 and NSUN6 knockout in HEK293T cells. The left panel illustrates the location of DRAM-induced mutation sites, which is highlighted in red asterisk. The right panel shows the corresponding quantification of Sanger sequencing (n=3).

The online version of this article includes the following source data and figure supplement(s) for figure 2:

**Source data 1.** PDF file containing the original western blots for **Figure 2E and F**, labeled with relevant bands and processing methods.

*Figure 2 continued on next page*

*Figure 2 continued*

**Source data 2.** Original western blots for *Figure 2E and F*.

**Figure supplement 1.** Expression of DRAM and DRAM[mut] in HEK293T cells.

**Figure supplement 2.** Editing of the DRAM system near the 5-methylcytosine (m⁵C) sites on 28S rRNA and tRNA.

**Figure supplement 3.** CRISPR/Cas9-mediated depletion of NSUN2 and NSUN6 in HEK293T.

**Figure supplement 3—source data 1.** PDF file containing original electrophoresis of the PCR products for *Figure 2—figure supplement 3B*, indicating the relevant bands and treatments.

**Figure supplement 3—source data 2.** Individual raw electrophoresis images of the PCR products for *Figure 2—figure supplement 3B*.

## DRAM enables transcriptome-wide analysis of m⁵C methylation

Subsequently, we performed RNA-seq analysis after DRAM transfection by detecting C-to-U/A-to-G editing events to accomplish transcriptome-wide detection of m⁵C (*Figure 3A*). To serve as positive controls, two previously published BS-seq datasets were also integrated (*Yang et al., 2017*; *Huang et al., 2019*). Mutations were detected near the m⁵C site in RPSA as A-to-G by DRAM-ABE (*Figure 3B*), and DRAM-CBE detected the presence of C-to-U mutations near the AP5Z1 m⁵C site (*Figure 3C*). However, the DRAM[mut] and Deaminase systems induced few effective mutations close to these sites. Examination of multiple reported high-confidence RNA m⁵C sites showed that DRAM-seq editing events were also enriched in the vicinity of the BS-seq sites (*Figure 3B and C*, *Figure 3—figure supplement 1*).

DRAM-seq analysis further confirmed that mutations in AP5Z1 and RPSA mRNA were reduced in methyltransferase knockout cells compared to wild-type cells (*Figure 3D and E*). Moreover, the knockout cells exhibited overall rare DRAM-seq editing events close to m⁵C sites in other mRNAs (*Figure 3—figure supplement 2*). These indicated that DRAM-seq analysis was detected in an m⁵C-dependent manner. Unfortunately, motif analysis failed to identify any sequence preferences or consensus motifs associated with DRAM-edited sites mediated by loci associated with NSUN2 or NSUN6 (*Figure 2—figure supplement 3D*).

A comparison of three biological replicates from each experimental group revealed a strong reproducibility of A-to-G/C-to-U mutations in HEK293T cells expressing DRAM-ABE and DRAM-CBE (*Figure 3—figure supplement 3*). Moreover, the DRAM-edited mRNAs revealed a high degree of overlap across the three biological replicates (*Figure 2—figure supplement 3C*). And a recent study by Wang et al. showed that ALYREF deletion affects the expression of 94 mRNAs (*Wang et al., 2023*), and only 55.32% of these ALYREF-regulated mRNAs can be detected by the DRAM system (*Figure 2—figure supplement 3E*). These findings suggest that DRAM selectively targets specific RNAs for editing, exhibiting a high degree of consistency across samples.

To obtain information on a set of high-confidence DRAM-seq data, we filtered the list of sites transfected with deaminase alone and screened the sequencing results with methyltransferase depleted, pooled editing events occurring in at least 10% of reads across multiple samples to obtain a set of high-confidence editing sites (*Figure 3F* and *Supplementary file 2*), and integrated genes with editing sites occurring in DRAM-ABE and DRAM-CBE (*Figure 3F* and *Supplementary file 3*).

Previous studies have indicated that m⁵C sites are predominantly distributed in the coding sequences (CDS) and notably enriched near the initiation codon (*Yang et al., 2017*; *Cui et al., 2017*; *Edelheit et al., 2013*; *Tang et al., 2020*; *Squires et al., 2012*; *Zhang et al., 2020a*). To further delineate the characteristics of the DRAM-seq data, we compared the distribution of DRAM-seq editing sites within the gene structure, specifically examining their occurrences in the 5'untranslated region (5'UTR), 3' untranslated region (3'UTR), CDS, and Intergenic/Intron region. Our analysis revealed that DRAM-seq editing events in cells expressing DRAM-ABE and DRAM-CBE were primarily located in the CDS and 3'UTR, indicating a nonrandom distribution of m⁵C (*Figure 3G*, *Figure 3—figure supplement 4A and B*). Moreover, plotting the distribution of DRAM-seq editing sites in mRNA segments (5'UTR, CDS, and 3'UTR) highlighted a significant enrichment in the CDS (*Figure 3H*). In contrast, cells expressing the deaminase exhibited a distinct distribution pattern of editing sites, characterized by a prevalence throughout the 3'UTR (*Figure 3H*). This finding reaffirms that the specific editing pattern observed in DRAM-seq across the transcriptome depends on its capacity to bind m⁵C.

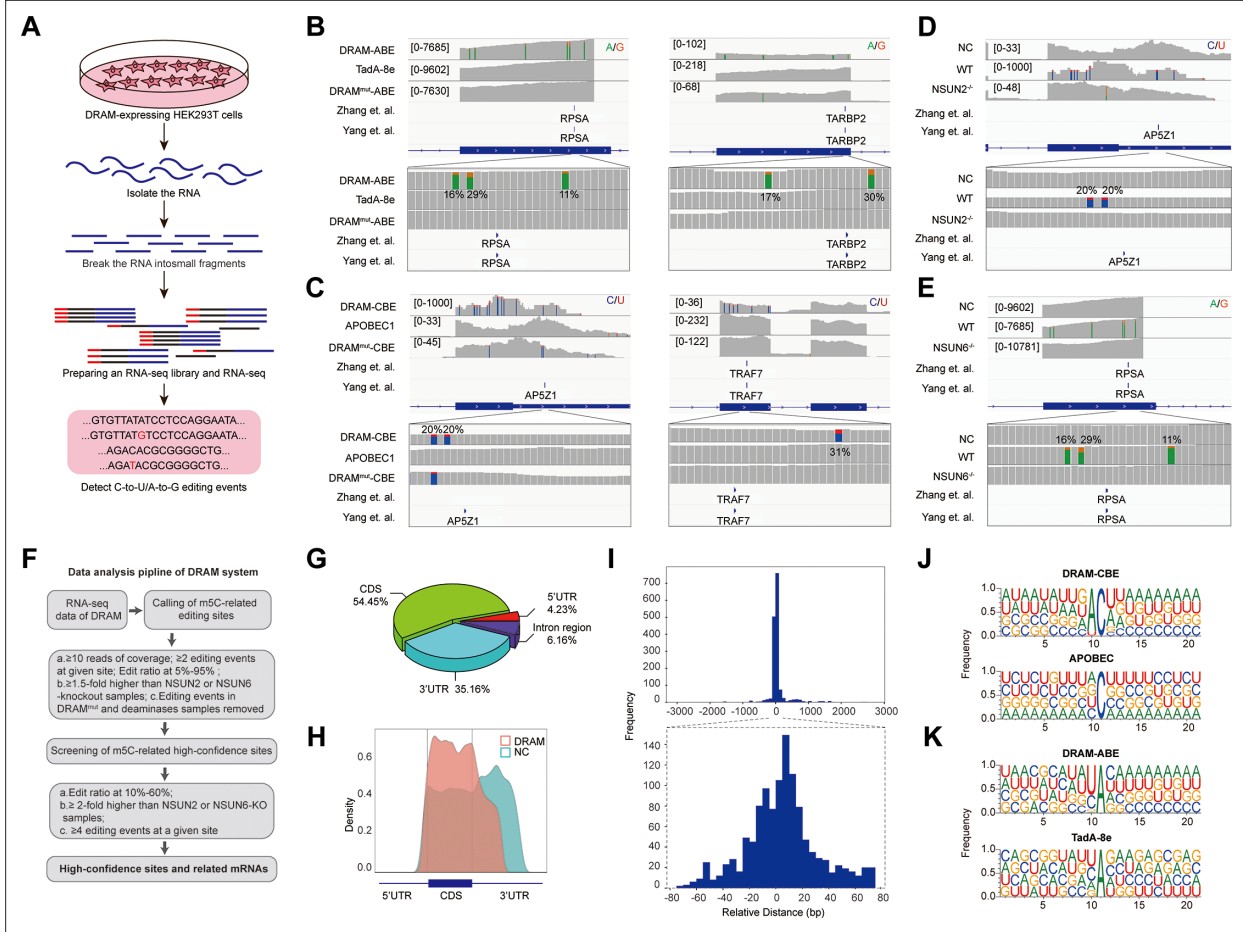

**Figure 3.** DRAM enables transcriptome-wide analysis of 5-methylcytosine (m⁵C) methylation. (**A**) Schematic of the DRAM-seq method. (**B, C**) Integrative genomics viewer (IGV) browser traces of DRAM-seq data expressing the indicated constructs in RPSA (**B**, left panel), TARBP2 (**B**, right panel), AP5Z1 (**C**, left panel), and TRAF7 (**C**, left panel) mRNAs. C-to-U or A-to-G mutations found in at least 10% of reads are indicated by coloring. The previously published RNA bisulfite-sequencing (BS-seq) datasets from two individual studies were displayed as panel 'Yang et al.' and 'Zhang et al.'. (n(DRAM)=3 independent samples, n(Deaminase)=2 independent samples, and n(DRAM^mut)=1 independent sample.) (**D, E**) IGV browser traces of DRAM-seq data in wild-type and methyltransferases knockout cells in AP5Z1 (**D**) and RPSA (**E**) mRNAs. C-to-U or A-to-G mutations that were found in at least 10% of reads are indicated by coloring. The previously published RNA BS-seq datasets from two individual studies were displayed as panel 'Yang et al.' and 'Zhang et al.'. n=3 independent samples. (**F**) Screening process for DRAM-seq assays and principles for screening high-confidence genes. (**G**) The pie chart shows the distribution of editing sites in different transcript region in cells expressing DRAM (n=3 independent samples). (**H**) The density map showing the distribution of editing events across the mRNA transcripts detected by DRAM-seq. (**I**) The frequency plot shows the distribution of the distances of edit events in DRAM-seq relative to the m⁵C sites from the published BS-seq datasets. The position of each m⁵C site of BS-seq is. determined as 0, and the relative distance of each site to the nearest edit event in DRAM-seq is calculated and plotted. The plots are presented separately based on the cutoff of upstream and downstream 3000 bp (above) and 80 bp (below) windows. (**J, K**) Motif analysis discovered within the ±20 nt region around the C-to-U or A-to-G editing site in cells expressing DRAM-CBE (**J**), APOBEC1(**J**), DRAM-ABE (**K**), and TadA-8e (**K**).

The online version of this article includes the following figure supplement(s) for figure 3:

**Figure supplement 1.** DRAM-seq editing in cellular RNAs.

**Figure supplement 2.** DRAM-seq analysis in NSUN2-depleted and NSUN6-depleted cellular RNAs.

**Figure supplement 3.** Analysis of DRAM-seq data repeatability.

**Figure supplement 4.** Transcriptome-wide mapping of 5-methylcytosine (m⁵C) in HEK293T cells.

Comparative analysis of the DRAM-seq editing sites with the previously published BS-seq m⁵C sites indicated that the likelihood of editing was notably higher in closer proximity to the m⁵C sites (*Figure 3I*). Furthermore, the editing window of DRAM exhibited enrichment approximately 20 bp before and after the m⁵C site (*Figure 3I*). Investigation into the sequences surrounding the editing window revealed that AC motifs were the most significantly enriched in DRAM-CBE, whereas (U/C)

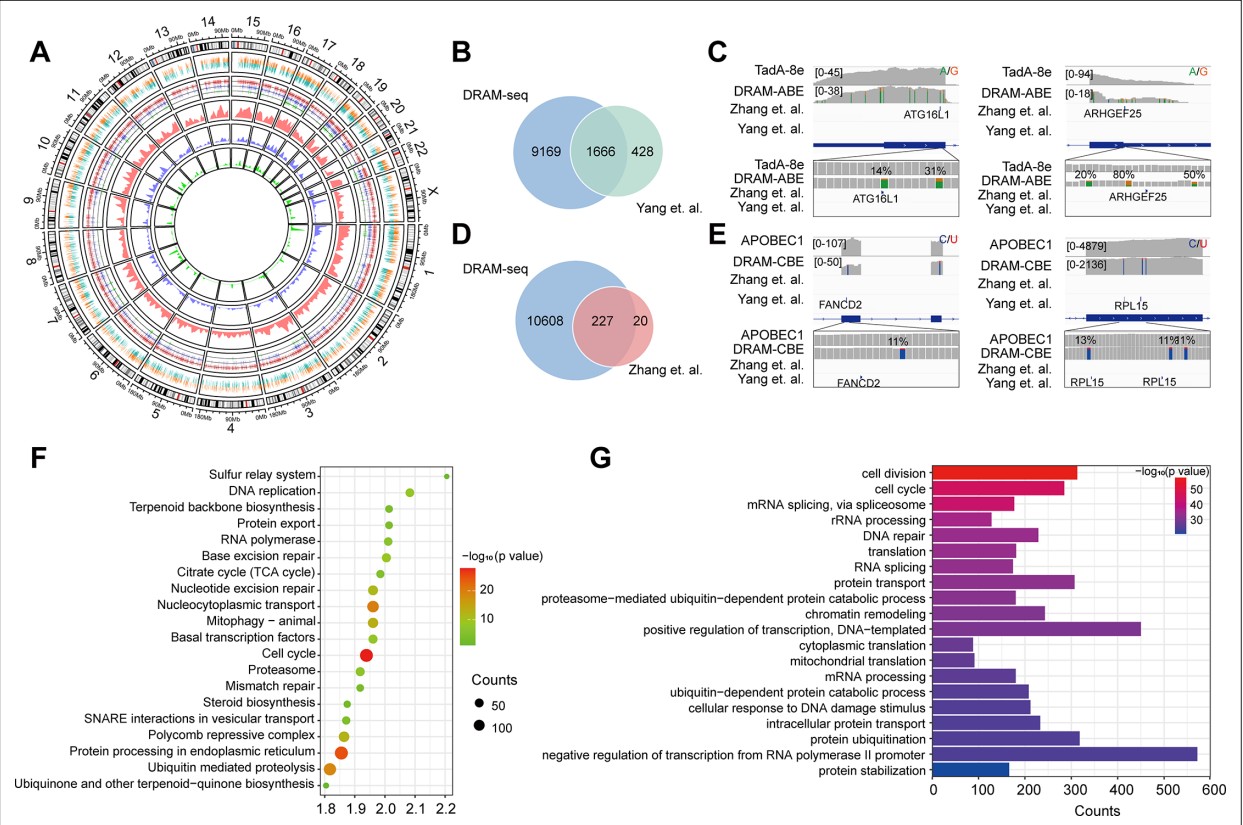

**Figure 4.** Stable and comprehensive cellular identification of 5-methylcytosine (m5C) loci by DRAM-seq. (**A**) Comparison of the overall distribution of genes with m5C modifications detected by DRAM-seq, Yang et al., and Zhang et al. on chromosomes. The mutation sites detected by DRAM-seq on each gene are categorized into dual-colored short lines, with positive strand mutations shown in orange and negative strand mutations in dark green. The line graph and kernel density plot in the inner ring represent the locations and distributions of overlapping genes detected by DRAM-seq (red), Yang et al. (blue), and Zhang et al. (light green). (**B**) Venn diagram showing the overlap between DRAM-seq and Yang et al.'s edited genes. (**D**) Venn diagram showing the overlap between DRAM-seq and Zhang et al.'s edited genes. (**C, E**) Integrative genomics viewer (IGV) browser traces of DRAM-seq data expressing the indicated constructs in the ATG16L1 (**B**), ARHGEF25 (**B**), FANCD2 (**D**), and RPL15 (**D**) mRNAs. C-to-U/A-to-G mutations found in at least 10% of reads are indicated by coloring, and the m5C site found by bisulfite-sequencing (BS-seq) is also labeled. (**F**) Genes with DRAM-seq editing events were analyzed for Kyoto Encyclopedia of Genes and Genomes (KEGG) bioprocess enrichment. (**G**) Gene Ontology (GO) biological processes enrichment analysis of genes with DRAM-seq editing events. Statistical analyses were performed using the DAVID tool.

A motifs were most notably enriched in DRAM-ABE. In contrast, the APOBEC1 and TadA-8e samples displayed no significantly enriched motifs, with mutations being more randomly orientated (*Figure 3J and K*).

## DRAM-seq provides stable and comprehensive identification of m5C loci

Subsequently, we then evaluated the ability of DRAM-seq to detect m5C across the entire transcriptome and compared its performance to that of the previously reported BS-seq. Although both previous studies employed bisulfite treatment, the resulting data obtained significant discrepancies due to variations in their treatment and analysis methodologies. We first complied the overall distribution of mutant regions identified by DRAM-seq, presenting both the mutant sites detected by the DRAM system and those reported in previous studies (*Yang et al., 2017*; *Chen et al., 2019a*) across each chromosome (*Figure 4A*). Our results indicated that DRAM-seq identified the presence of m5C modifications covering 79.6% of the genes detected by *Yang et al., 2017*, and 91.9% of the genes detected by *Huang et al., 2019* (*Figure 4B and D*). Remarkably, certain pivotal regulators with diverse biological functions, such as ATG16L1 (coordinates autophagy pathway) (*Fletcher et al., 2018*) and ARHGEF25 (plays an important role in actin cytoskeleton reorganization) (*Guo et al., 2003*), were identified by Zhang et al. and DRAM-seq, but not by Yang et al. (*Figure 4C*). Conversely,

FANCD2 (maintains chromosome stability) (*Naim and Rosselli, 2009*) and RPL15 (components of the large ribosomal subunit) (*Anger et al., 2013*; *Liang et al., 2020*) were discovered by Yang et al. and DRAM-seq, but not by Zhang et al. (*Figure 4E*). Hence, DRAM-seq appears to offer a more stable and comprehensive identification of the m⁵C loci.

To provide functional insights into m⁵C RNA-modified genes in HEK293T cells, we conducted Gene Ontology (GO) and Kyoto Encyclopedia of Genes and Genomes (KEGG) analyses. These results highlighted the involvement of these genes in the regulation of diverse key biological processes, such as cell division, cell cycle, mRNA splicing, protein processing in the endoplasmic reticulum, nucleocytoplasmic transport, translation, DNA repair, and others (*Figure 4F and G*, *Figure 3—figure supplement 4C and D*).

## DRAM enables low-input m⁵C profiling

A significant challenge in m⁵C detection lies in the specificity of antibodies and the substantial amount of input RNA required for sequencing. RNA is susceptible to degradation during denaturation, sodium bisulfite treatment, and desulfurization steps in the BS-seq assay (*Schaefer, 2015*). Immunoprecipitation-based m⁵C assays and LC-MS/MS also impose high demand for sample input (*Hussain et al., 2013*; *Cui et al., 2017*; *Bourgeois et al., 2015*). Several experiments have highlighted the requirement of 100–500 ng of RNA for m⁵C-RIP-seq, while BS-seq necessitates an even more demanding 750–1000 ng of RNA (*Huang et al., 2019*; *Cui et al., 2017*; *Gu and Liang, 1933*). To assess the detection limits of DRAM-Sanger, we attempted to amplify two representative m⁵C-containing sites in the RPSA and AP5Z1 transcripts from diluted RNA samples.

Remarkably, we successfully generated PCR products of these two mRNAs from cDNAs corresponding to 250 ng, 50 ng, and 10 ng of total RNA. Quantitative analysis by Sanger sequencing demonstrated nearly identical Sanger traces across these dilutions (*Figure 5A and B*). This finding underscores that the specificity of DRAM editing depended on its ability to bind m⁵C, and DRAM is proficient in low-input m⁵C analyses. Furthermore, cell viability was determined by CCK8 assay on HEK293T cells transfected with DRAM (*Figure 5C*). Importantly, there was no significant difference in the relative proliferative capacity of the cells compared to untransfected cells (NC), indicating that DRAM expression did not adversely affect cell viability (*Figure 5D*).

Transfection of the DRAM system in cells results in the transient overexpression of fusion proteins. To investigate how varying expression levels of these proteins influence A-to-G and C-to-U editing within the same m⁵C region, we conducted a gradient transfection using plasmid concentrations of 1500 ng, 1000 ng, and 500 ng. This approach allowed us to progressively reduce the expression levels of the fusion proteins (*Figure 5E and F*). Sanger sequencing revealed that the editing efficiency of A-to-G and C-to-U within the m⁵C region significantly decreased as fusion protein expression diminished (*Figure 5G and H*). These findings suggest that the transfection efficiency of the DRAM system is concentration-dependent and that the ratio of editing efficiency to transfection efficiency may assist in the quantitative analysis of m⁵C using the DRAM system.

## Discussion

In recent years, m⁵C methylation modifications have received increasing attention, with multiple reports detailing the distribution of RNA m⁵C methylation modifications across various species and tissues, elucidating their characteristics. Despite the relatively low abundance of m⁵C, its highly dynamic changes hold significant implications for the regulation of physiological and pathological processes (*Yang et al., 2017*; *Huang et al., 2019*; *Liu et al., 2021*). However, due to the limitations of sequencing methods and the variability of data processing, there remains ample room for progress in the study of m⁵C detection methods.

In this study, we developed a site-specific, depth-sequencing-free m⁵C detection method using DRAM-Sanger. This workflow relies on conventional molecular biology assays such as RT-PCR and Sanger sequencing, eliminating the need for specialized techniques and thereby simplifying the process of m⁵C detection.

DRAM-seq introduces a novel strategy for transcriptome-wide m⁵C detection, overcoming inherent limitations in existing methods. Notably, DRAM-seq covered around 80% of the high-confidence m⁵C-modified genes detected by BS-seq and identified more potential m⁵C sites. This can be attributed

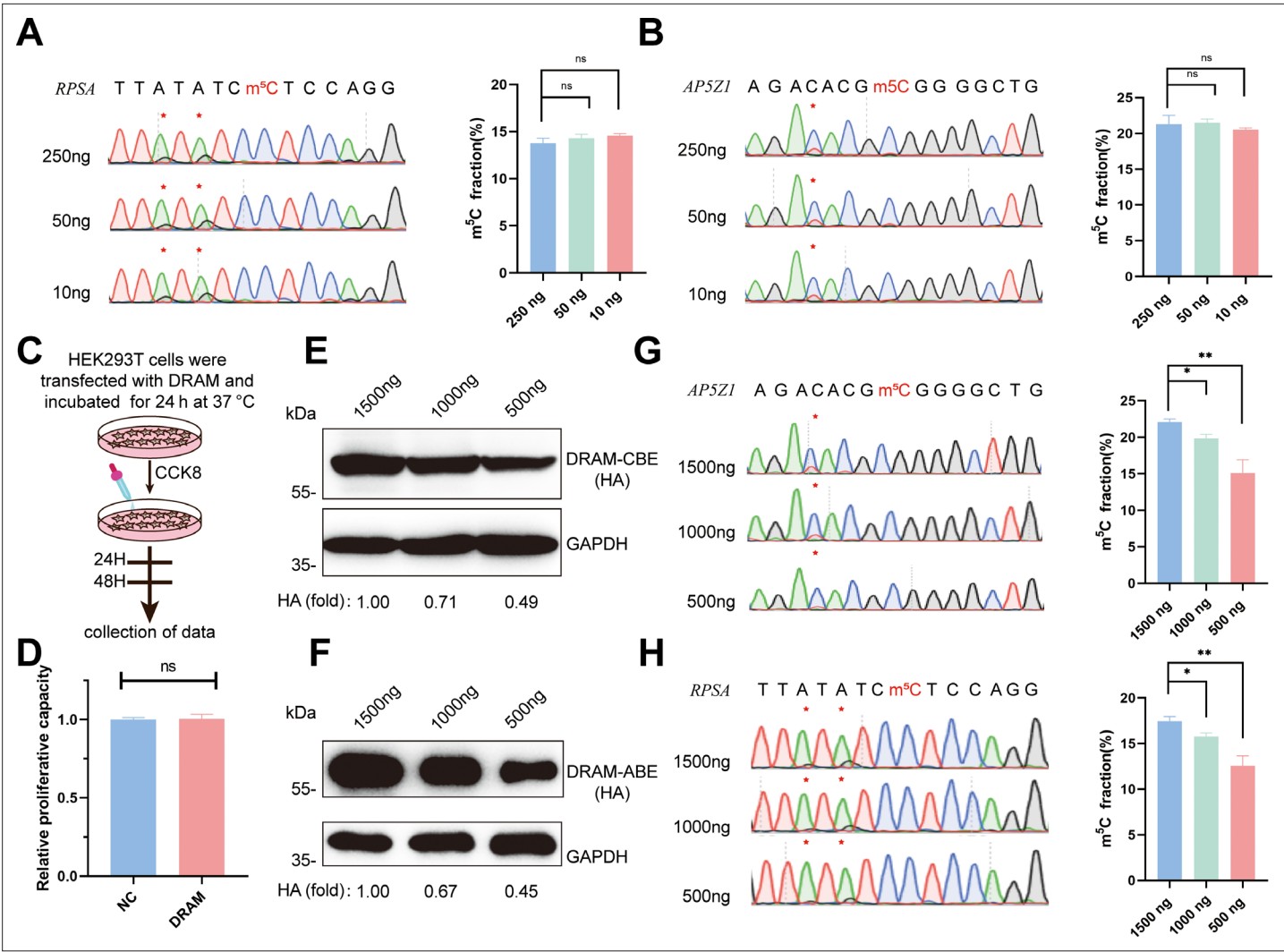

**Figure 5.** Low-input 5-methylcytosine (m5C) detection and transfection efficiency of DRAM system. (**A, B**) DRAM analysis of RPSA (**A**) and AP5Z1 (**B**) mRNAs with 250 ng, 50 ng, and 10 ng of input RNA. Representative Sanger sequencing plots are shown on the left panel, with mutation sites marked with asterisks. The mutation rates are quantified on the right panel. (**C**) Flowchart illustrating cell viability analysis by CCK8 reagent after DRAM transfection in HEK293T cells. (**D**) Quantitative comparison of the relative proliferative capacity of DRAM-expressing and untransfected cells. (**E, F**) The expression levels of DRAM-CBE (**E**) and DRAM-ABE (**F**) systems at different plasmid transfection concentrations were verified by western blotting. (**G, H**) Editing of RPSA (**G**) and AP5Z1 (**H**) mRNA at varying concentrations of DRAM protein expression. The left panels indicate Sanger sequencing results following RT-PCR, while the corresponding quantifications of DRAM-induced mutations are shown in the right panels.

The online version of this article includes the following source data for figure 5:

**Source data 1.** PDF file containing the original western blots for *Figure 5E-H*, labeled with relevant bands and processing methods.

**Source data 2.** Original western blots for *Figure 5E-H*.

to the avoidance of bisulfite treatment by DRAM-seq, preventing RNA damage and ensuring a more comprehensive representation of RNA samples. This feature also likely contributes to the observed stability of DRAM-seq in comparison to BS-seq. Additionally, DRAM-seq is not limited by antibody specificity and is resistant to chemical-induced damage.

A prominent challenge in existing m5C profiling methods is their reliance on substantial amounts of input RNA samples. In contrast, DRAM operates through the deamination activity of deaminase, preserving RNA integrity and preventing degradation. The notable advantage of DRAM lies in its capacity for low-input m5C detection. Our analysis demonstrates that DRAM requires as low as 10 ng of total RNA for m5C detection. While DRAM is currently well suited for detecting m5C on a transcriptome-wide scale, the potential for future applications involving third-generation sequencing

could extend its utility to individual mRNAs, particularly m5C heterogeneity on mRNA splicing variants. In addition, the DRAM system depends on the specific recognition of m5C modifications on ssRNA by the reader protein, theoretically avoiding the false-positive effects of 5-hydroxymethylation modifications in other assays, such as BS-seq (*Huang et al., 2019*; *Trixl et al., 2019*; *Schaefer et al., 2009*). This potential feature could enhance the accuracy of the DRAM assay, albeit it still requires careful validation.

In our study, m5C detection was performed following the transient transfection of the DRAM detection system into mammalian cells, which might result in a lower mutation rate at the corresponding site. Therefore, employing lentiviral-mediated transfection into cell lines of interest could potentially enhance the efficiency of m5C detection. Our results confirm that YBX1 and ALYREF exhibit specificity as m5C readers, binding preferentially to RNAs with m5C modifications, thereby validating the reliability of the DRAM detection system. However, mutations in the key amino acids responsible for m5C binding reduced their affinity while retaining some binding capacity. DRAM-seq analysis identified a substantial number of m5C sites. However, we cannot exclude the potential existence of false-positive sites resulting from nonspecific binding of the m5C reader. Further elucidation of the key amino acids directing ALYREF and YBX1's binding to m5C methylation sites should enable more accurate and sensitive m5C detection by DRAM-seq. Due to the lack of a fixed base composition for characterizing all m5C modification sites, DRAM has an apparent limitation in achieving single-base resolution for detecting m5C. This technical constraint may explain the absence of identifiable sequence specificity in our analysis of m5C sites catalyzed by NSUN2 and NSUN6, despite previous reports associating these methyltransferases with 'G'-rich sequences and the 'CUCCA' motif (*Selmi et al., 2021*). However, our present study proved that the measuring resolution of DRAM is around 40 nt, which facilitates higher precision than that of m5C-RIP-seq (~100 nt). In the future, with more in-depth analyses of m5C reader structures and the identification of new potential m5C readers, we expect to achieve more precise m5C localization and more comprehensive m5C modification detection. Moreover, the substitution of deaminases, such as A3A and A3G (the family members of APOBEC), could also potentially enhance the efficiency of the DRAM detection (*Kim et al., 2023*; *Barka et al., 2022*; *Pan et al., 2017*).

Although the m5C assay can be performed using the DRAM system alone, comparing it with the DRAM^mut and deaminase controls could enhance the accuracy of m5C detection in specific regions. Given that the expression of DRAM fusion proteins significantly influences m5C detection, it is advisable to transfect the same batch of cells during the assay to ensure consistent transfection efficiency across experimental groups and thus can better standardize the detection.

One future direction of endeavor is the purification of DRAM fusion proteins to facilitate in vitro detection of RNA m5C methylation, which could extend the scope of DRAM-seq to diverse sample types. Another potential application for DRAM-seq could be the expression of drug-inducible DRAM systems in vivo using various animal models for m5C analysis. These will together provide novel insights into m5C modifications for biological and clinical research.

## Conclusions

In summary, we developed a novel deaminase and reader protein-assisted RNA m5C methylation approach that detects the m5C region by deaminating As or Cs in close proximity to the m5C sites, which does not rely on antibodies or bisulfite, thus leading to unprecedently comprehensive transcriptome-wide RNA m5C methylation profiling. We anticipated that this system could pave the way for uncovering further biological functions of m5C modifications and facilitate the development of therapeutic interventions for associated diseases.

## Materials and methods
### Plasmid construction

ALYREF and YBX1 expression plasmids were purchased from MIAOLING BIOLOGY (http://www.miaolingbio.com/), and the ALYREF and YBX1 fractions were then amplified using specific primer. The ALYREF and YBX1 portions were amplified using pCMV-APOBEC1-YTH (RRID:Addgene_131636) and ABE8e (RRID:Addgene_138489) to amplify the deaminase portion and the essential plasmid construct proxies, and finally the fragments were recombined by the ClonExpress Ultra One Step Cloning Kit to complete the plasmid vector construction. Both DRAM^mut-ABE- and DRAM^mut-CBE-related vectors

were obtained by introducing the corresponding key amino acid mutations using Fast Site-Directed Mutagenesis Kit (TIANGEN Biotech). The primer sequences used are listed in *Supplementary file 4*.

## Cell culture and plasmid transfection

HEK293T cell line (ATCC) was cultured in Dulbecco's Modified Eagle Medium (DMEM) supplemented with 10% fetal bovine serum (CLARK BIOSCIENCE) and 1% penicillin (100 U/ml)-streptomycin (100 μg/ml). The cells were seeded in 12-well plates and transfected using Hieff Trans Liposomal Transfection Reagent (Yeasen).

NSUN2-depleted cell lines were generated by cloning NSUN2-targeting single guide RNA sequences into the pSpCas9(BB)-2A-Puro (PX459) V2.0 plasmid (RRID:Addgene_62988). Plasmids were then transfected into HEK293T cells and Puromycin (Meilunbio) was added at a final concentration of 3 μg/ml to enrich the positively transfected cells 24 hr after transfection. After 72 hr, the cells were collected and used for genotyping by Sanger sequencing. NSUN6-depleted cell lines were generated in the same way. The primers used for genotyping and single guide RNA sequences are listed in *Supplementary file 4*.

## Cell viability measurements

HEK293T cells were transfected with DRAM plasmid and cultured at 37°C for 24 hr. Subsequently, 1000 cells were seeded in 96-well plates. After waiting for the cells to attach to the wall, the cell activity was detected by Cell Counting Kit-8 (Meilunbio). Cell Counting Kit-8 contains WST-8, which in the presence of the electronically coupled reagent 1-Methoxy PMS can be reduced by mitochondrial dehydrogenase to the orange-colored metazan product Formazan, the absorbance of which is measured at 450 nm to analyze cellular activity.

## Western blotting

For protein blotting, samples were lysed in RIPA Lysis Buffer (Meilunbio) with phenylmethanesulfonyl fluoride, and the BCA protein assay kit (Beyotime Biotechnology) was used to measure protein concentration. Total protein extracts were separated by SDS-PAGE on a 10% gel and then transferred to 0. 45 nm polyvinylidene fluoride membranes (Boster). Subsequently, the proteins were probed with specific antibodies after the blot was blocked with 5% nonfat milk (Boster). Images were quantified using ImageJ software and all data are expressed as mean ± SEM.

The following antibodies and concentrations were used: NSUN2 Polyclonal antibody (Proteintech; Cat No. 20854-1-AP; 1:7500), NSUN6 Polyclonal antibody (Proteintech; Cat No. 17240-1-AP; 1:2000), RabbitAnti-GAPDH antibody (Bioss; bs-41373R; 1:2000), Alpha Tubulin Polyclonal antibody (Proteintech; Cat No. 11224-1-AP; 1:2000), HRP-labeled Goat Anti-Rabbit IgG(H+L) (Beyotime Biotechnology; A0208; 1:2000).

## cDNA synthesis and Sanger sequencing

Total cellular RNA was extracted with TRIzol reagent (TIANGEN Biotech) and cDNA was synthesized using PrimeScript II 1st Strand cDNA Synthesis Kit (Takara Bio) according to the manufacturer's recommendations. PCR was then performed using 2×Taq PCR MasterMix II (TIANGEN Biotech) and primers flanking $m^5C$ target sites, and the purified PCR products were directly sequenced by Sanger sequencing. The Sanger sequencing results were analyzed using EditR 1.0.10 to calculate the mutation frequency (*Kluesner et al., 2018*). The primers used in this study are shown in *Supplementary file 4*.

## Real-time quantitative PCR

cDNA was synthesized using FastKing RT kit (with gDNase) (TIANGEN Biotech) according to the manufacturer's recommendations. RT-qPCR assay was performed using SuperReal PreMix Plus (SYBR Green) (TIANGEN Biotech). GAPDH was used as an endogenous control, and the expression levels were normalized to the control and calculated by the $2^{-\Delta\Delta Ct}$ formula. All samples were analyzed in triplicate and each mRNA quantification represents the average of at least three measurements. All data are expressed as mean ± SEM. The primers used in this study are shown in *Supplementary file 4*.

## Protein structure modeling

Protein structure simulations were performed using the SWISS-MODEL online website (https://swiss-model.expasy.org/interactive) (*Bienert et al., 2017*). The SWISS-MODEL database is able to provide up-to-date annotated 3D protein models, which are generated from automated homology modeling of related model organisms and experimental structural information for all sequences in UniProtKB, with reliable structural information, and subsequently protein structure observations were performed using PyMOL (*Delano, 2002*).

## BS-seq PCR

We referenced BS-seq PCR, an assay established by Schaefer et al. We chemically deaminated cytosine in RNA using the EZ RNA methylation kit (50) (Zymo Research) and then quantified m5C methylation levels based on PCR amplification of cDNA combined with deep sequencing (*Schaefer et al., 2009*).

RNA Conversion Reagent was premixed with prepared RNA samples, and the RNA was denatured at 70°C for 5 min, followed by a reaction period of 45 min at 54°C. Finally, the purified RNA samples were recovered after desulfurization by RNA Desulphonation Buffer. cDNA was synthesized using PrimeScript II 1st Strand cDNA Synthesis Kit (Takara Bio) according to the manufacturer's recommendations. PCR was then performed using 2×EpiArt HS Taq Master Mix (Dye Plus) (Vazyme) and m5C target site-specific Bisulfite Primer (primer sequences were designed at https://zymoresearch.eu/pages/bisulfite-primer-seeker), the products were purified by TIANgel Midi Purification Kit (TIANGEN Biotech), and the connectors for second-generation sequencing were attached at both ends of the products for sequencing. Finally, deep sequencing was performed by HiTOM analysis to detect the methylation level (the number of reads>1000 in deep sequencing) (*Liu et al., 2019*). The primers used in this study are shown in *Supplementary file 4*.

## Library construction and next-generation sequencing

1 μg of total cellular RNA was used for sequencing library generation by NEBNext Ultra RNA Library Prep Kit for Illumina (NEB, USA, Catalog #: E7530L) following the manufacturer's recommendations and index codes were added to attribute sequences to each sample. Briefly, mRNA was purified from total RNA using poly-T oligo-attached magnetic beads. Fragmentation was carried out using divalent cations under elevated temperature in NEB Next First Strand Synthesis Reaction Buffer(5×). First-strand cDNA was synthesized using random hexamer primer and M-MuLV Reverse Transcriptase (RNase H). Second-strand cDNA synthesis was subsequently performed using DNA Polymerase I and RNase H. Remaining overhangs were converted into blunt ends via exonuclease/polymerase activities. After adenylation of 3′ ends of DNA fragments, NEB Next Adaptor with hairpin loop structure was ligated to prepare for hybridization. To select cDNA fragments of preferentially 370–420 bp in length, the library fragments were purified with AMPure XP system (Beverly, USA). Then, 3 μl USER Enzyme (NEB, USA) was used with size-selected, adaptor-ligated cDNA at 37°C for 15 min followed by 5 min at 95°C before PCR. Then PCR was performed with Phusion High-Fidelity DNA polymerase, Universal PCR primers, and Index (X) Primer. At last, PCR products were purified (AMPure XP system) and library quality was assessed on the Agilent 5400 system (Agilent, USA) and quantified by qPCR (library concentration≥1.5 nM). The qualified libraries were pooled and sequenced on Illumina platforms with PE150 strategy in Novogene Bioinformatics Technology Co., Ltd (Beijing, China), according to effective library concentration and data amount required.

## DRAM-seq analysis and calling of edited sites

The raw fastq sequencing data were cleaned by trimming the adapter sequences using Fastp (v0.23.1) and were aligned to the human genome (hg19) using STAR (v2.7.7) in paired-end mode. The aligned BAM files were sorted and PCR duplicates were removed using Samtools (v1.12). The cite calling of DRAM-seq was performed using Bullseye, a previously customized pipeline to look for C-to-U or A-to-G edited sites throughout the transcriptome (*Meyer, 2019*). Briefly, the sorted and deduplicated BAM files were initially parsed by parseBAM.pl script.

Then, Find_edit_site.pl script was employed to find C-to-U or A-to-G editing events by DRAM-seq with at least 10 reads of coverage, an edit ratio of 5–95%, an edit ratio at least 1.5-fold higher than NSUN2 or NSUN6-knockout samples, and at least two editing events at a given site. Sites that were

only found in one replicate of each DRAM protein variant were removed. Editing events appeared in cells expressing merely APOBEC1 or TadA8e were also removed. For high-confidence filtering, we further adjusted the Find_edit_site.pl parameters to the edit ratio of 10–60%, an edit ratio of control samples at least twofold higher than NSUN2 or NSUN6-knockout samples, and at least four editing events at a given site.

## Metagene and motif analyses

Metagene analysis was performed using hg19 annotations according to previously reported tool, MetaplotR (*Olarerin-George and Jaffrey, 2017*). For motif analysis, the 20 bp flanking sequence of each DRAM-seq editing site was extracted by Bedtools (v2.30.0) (*Quinlan and Hall, 2010*). The motif logos were then plotted by WebLogo (v3.7.12) (*Crooks et al., 2004*).

## Replicates analysis

Independent biological replicates of DRAM-ABE or DRAM-CBE in DRAM-seq analysis were separately compared by computing the Pearson correlation coefficient between the number of C-to-U mutations per mRNA between any two replicate experiments.

## GO and KEGG analysis

GO and KEGG analysis of DRAM-seq edited mRNAs was performed using the DAVID bioinformatic database (*Sherman et al., 2022*). GO terms with a p value of less than 0.05 were considered statistically significant.

## RNA pull-down assay

The biotin-labeled RNA oligonucleotides with (oligo-$m^5C$) or without $m^5C$ (oligo-C) were prepared in advance: 5'-biotin-GAGGUAUGAAXUGUAAGTT-3' (X=C or $m^5C$, used in the ALYREF and ALYREF[mut] group) and 5'-biotin-GAAAGGAGAUXGCCAUUAUCC-3' (X=C or $m^5C$, used in the YBX1 and YBX1[mut] group). Protein lysates were then isolated from HEK293T cells transfected with DRAM-YBX1, DRAM-YBX1[mut], DRAM-ALYREF, or DRAM-ALYREF[mut] for 24 hr using lysis buffer. RNA pull-down assays were performed with the Pierce Magnetic RNA-Protein Pull-Down Kit (Thermo) following the manufacturer's instructions, and the results were finally analyzed by western blotting.

## Statistical analysis

All data are expressed as mean ± SEM of three independent determinations. Data were analyzed through a two-tailed t-test. A probability of $p<0.05$ was considered statistically significant; *, $p<0.05$, **, $p<0.01$, ***, $p<0.001$, and ****, $p<0.0001$ denote the significance thresholds; ns denotes not significant.

## Acknowledgements

We thank Yuning Song, Yuanyuan Xu, and Tingting Sui for critical feedback on the work and manuscript. This work was supported by the National Natural Science Foundation of China (No. 32200466).

## Additional information

### Funding

| Funder | Grant reference number | Author |
| --- | --- | --- |
| National Natural Science Foundation of China | 32200466 | Yang Han |

The funders had no role in study design, data collection and interpretation, or the decision to submit the work for publication.

### Author contributions

Jiale Zhou, Conceptualization, Resources, Data curation, Software, Formal analysis, Supervision, Validation, Investigation, Visualization, Methodology, Writing – original draft, Project administration,

Writing – review and editing; Ding Zhao, Conceptualization, Resources, Data curation, Software, Formal analysis, Supervision, Validation, Investigation, Visualization, Methodology, Project administration; Jinze Li, Conceptualization, Resources, Data curation, Software, Formal analysis, Supervision, Validation, Investigation, Visualization, Methodology, Writing – original draft, Project administration; Deqiang Kong, Xiangrui Li, Renquan Zhang, Yuru Liang, Supervision, Investigation; Xun Gao, Yuqiang Qian, Di Wang, Jiahui Chen, Supervision, Visualization; Liangxue Lai, Zhanjun Li, Conceptualization, Resources, Data curation, Software, Formal analysis, Supervision, Funding acquisition, Validation, Investigation, Visualization, Methodology, Project administration, Writing – review and editing; Yang Han, Conceptualization, Resources, Data curation, Software, Formal analysis, Supervision, Funding acquisition, Validation, Investigation, Visualization, Methodology, Writing – original draft, Project administration, Writing – review and editing

### Author ORCIDs
Jiale Zhou https://orcid.org/0009-0006-9199-8647
Liangxue Lai https://orcid.org/0000-0002-5256-6802
Yang Han https://orcid.org/0000-0001-6249-370X
Zhanjun Li https://orcid.org/0000-0001-6914-8589

Reviewer #2 (Public review): https://doi.org/10.7554/eLife.98166.5.sa1
Author response https://doi.org/10.7554/eLife.98166.5.sa2

## Additional files

### Supplementary files
Supplementary file 1. The detection for 5-methylcytosine ($m^5C$) in RNA.

Supplementary file 2. High-confidence editing sites of DRAM-seq.

Supplementary file 3. Genes with editing sites occurring in DRAM-seq. (a) DRAM-edited genes. (b) Overlapped genes with Yang et al. (c) Overlapped genes with Zhang et al.

Supplementary file 4. Primers and sgRNA for molecular cloning, genotyping, and downstream analyses. (a) The primers for vector construction. (b) The primers used for genotyping. (c) Single guide RNA. (d) The primers for DRAM-Sanger analysis. (e) The primers for qPCR. (f) The primers for bisulfite-sequencing PCR.

MDAR checklist

### Data availability
The data supporting the findings of this study have been deposited in Gene Expression Omnibus under accession GSE254194.

The following dataset was generated:

| Author(s) | Year | Dataset title | Dataset URL | Database and Identifier |
|---|---|---|---|---|
| Zhou J, Zhao D, Li J, Kong D, Li X, Zhang R, Liang Y, Gao X, Qian Y, Wang D, Chen J, Lai L, Han Y Li Z | 2025 | Transcriptome-wide identification of 5-methylcytosine by deaminase and reader protein-assisted sequencing | https://www.ncbi.nlm.nih.gov/geo/query/acc.cgi?acc=GSE254194 | NCBI Gene Expression Omnibus, GSE254194 |

The following previously published datasets were used:

| Author(s) | Year | Dataset title | Dataset URL | Database and Identifier |
|---|---|---|---|---|
| Rui Z | 2019 | Genome-wide identification of mRNA 5-methylcytosine in mammals | https://www.ncbi.nlm.nih.gov/geo/query/acc.cgi?acc=GSE122260 | NCBI Gene Expression Omnibus, GSE122260 |

*Continued on next page*

*Continued*

| Author(s) | Year | Dataset title | Dataset URL | Database and Identifier |
|---|---|---|---|---|
| Yang X, Yang Y, Sun BF, Chen YS | 2017 | 5-methylcytosine promotes mRNA export | https://www.ncbi.nlm.nih.gov/geo/query/acc.cgi?acc=GSE93751 | NCBI Gene Expression Omnibus, GSE93751 |

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
