## [Editor Report · eLife Assessment]

This potentially **useful** study introduces an orthogonal approach for detecting RNA modification, without chemical modification of RNA, which often results in RNA degradation and therefore loss of information. Compared to previous versions, the most recent one is improved and sufficiently aligned with the standards of the field to merit consideration by the research community, making the evidence **solid** according to said standards. Nevertheless, uncertainty regarding false positive and false negative rates remains, as it does for some of the alternative approaches. With more rigorous validation, the approach might be of particular interest for sites in RNA molecules where modifications are rare.

---

## [Referee Report · Reviewer #2 (Public review)]

The fledgling field of epitranscriptomics has encountered various technical roadblocks with implications as to the validity of early epitranscriptomics mapping data. As a prime example, the low specificity of (supposedly) modification-specific antibodies for the enrichment of modified RNAs, has been ignored for quite some time and is only now recognized for its dismal reproducibility (between different labs), which necessitates the development of alternative methods for modification detection. Furthermore, early attempts to map individual epitranscriptomes using sequencing-based techniques are largely characterized by the deliberate avoidance of orthogonal approaches aimed at confirming the existence of RNA modifications that have been originally identified.

Improved methodology, the inclusion of various controls, and better mapping algorithms as well as the application of robust statistics for the identification of false-positive RNA modification calls have allowed revisiting original (seminal) publications whose early mapping data allowed making hyperbolic claims about the number, localization and importance of RNA modifications, especially in mRNA. Besides the existence of m6A in mRNA, the detectable incidence of RNA modifications in mRNAs has drastically dropped.

As for m5C, the subject of the manuscript submitted by Zhou et al., its identification in mRNA goes back to Squires et al., 2012 reporting on >10.000 sites in mRNA of a human cancer cell line, followed by intermittent findings reporting on pretty much every number between 0 to > 100.000 m5C sites in different human cell-derived mRNA transcriptomes. The reason for such discrepancy is most likely of a technical nature. Importantly, all studies reporting on actual transcript numbers that were m5C-modified relied on RNA bisulfite sequencing, an NGS-based method, that can discriminate between methylated and non-methylated Cs after chemical deamination of C but not m5C. RNA bisulfite sequencing has a notoriously high background due to deamination artifacts, which occur largely due to incomplete denaturation of double-stranded regions (denaturing-resistant) of RNA molecules. Furthermore, m5C sites in mRNAs have now been mapped to regions that have not only sequence identity but also structural features of tRNAs. Various studies revealed that the highly conserved m5C RNA methyltransferases NSUN2 and NSUN6 do not only accept tRNAs but also other RNAs (including mRNAs) as methylation substrates, which in combination account for most of the RNA bisulfite-mapped m5C sites in human mRNA transcriptomes. Is m5C in mRNA only a result of the Star activity of tRNA or rRNA modification enzymes, or is their low stoichiometry biologically relevant?

In light of the short-comings of existing tools to robustly determine m5C in transcriptomes, other methods, like DRAM-seq, aiming to map m5C independently of ex situ RNA treatment with chemicals, are needed to arrive at a more solid "ground state", from which it will be possible to state and test various hypotheses as to the biological function of m5C, especially in lowly abundant RNAs such as mRNA.

Importantly, the identification of >10.000 sites containing m5C increases through DRAM-Seq, increases the number of potential m5C marks in human cancer cells from a couple of 100 (after rigorous post-hoc analysis of RNA bisulfite sequencing data) by orders of magnitude. This begs the question, whether or not the application of these editing tools results in editing artefacts overstating the number of actual m5C sites in the human cancer transcriptome.

[Editors' note: earlier reviews have been provided here: https://doi.org/10.7554/eLife.98166.3.sa1; https://doi.org/10.7554/eLife.98166.2.sa1; https://doi.org/10.7554/eLife.98166.1.sa1]

---

## [Author Response]

The following is the authors’ response to the original reviews.

**Responses to Reviewer’s Comments:**

**To Reviewer #2:**
(1) The use of two m^5^C reader proteins is likely a reason for the high number of edits introduced by the DRAM-Seq method. Both ALYREF and YBX1 are ubiquitous proteins with multiple roles in RNA metabolism including splicing and mRNA export. It is reasonable to assume that both ALYREF and YBX1 bind to many mRNAs that do not contain m^5^C.To substantiate the author's claim that ALYREF or YBX1 binds m^5^C-modified RNAs to an extent that would allow distinguishing its binding to non-modified RNAs from binding to m^5^Cmodified RNAs, it would be recommended to provide data on the affinity of these, supposedly proven, m^5^C readers to non-modified versus m^5^C-modified RNAs. To do so, this reviewer suggests performing experiments as described in Slama et al., 2020 (doi: 10.1016/j.ymeth.2018.10.020). However, using dot blots like in so many published studies to show modification of a specific antibody or protein binding, is insufficient as an argument because no antibody, nor protein, encounters nanograms to micrograms of a specific RNA identity in a cell. This issue remains a major caveat in all studies using so-called RNA modification reader proteins as bait for detecting RNA modifications in epitranscriptomics research. It becomes a pertinent problem if used as a platform for base editing similar to the work presented in this manuscript.The authors have tried to address the point made by this reviewer. However, rather than performing an experiment with recombinant ALYREF-fusions and m^5^C-modified to unmodified RNA oligos for testing the enrichment factor of ALYREF in vitro, the authors resorted to citing two manuscripts. One manuscript is cited by everybody when it comes to ALYREF as m^5^C reader, however none of the experiments have been repeated by another laboratory. The other manuscript is reporting on YBX1 binding to m^5^C-containing RNA and mentions PARCLiP experiments with ALYREF, the details of which are nowhere to be found in doi: 10.1038/s41556-019-0361-y.Furthermore, the authors have added RNA pull-down assays that should substitute for the requested experiments. Interestingly, Figure S1E shows that ALYREF binds equally well to unmodified and m^5^C-modified RNA oligos, which contradicts doi:10.1038/cr.2017.55, and supports the conclusion that wild-type ALYREF is not specific m^5^C binder. The necessity of including always an overexpression of ALYREF-mut in parallel DRAM experiments, makes the developed method better controlled but not easy to handle (expression differences of the plasmid-driven proteins etc.)

Thank you for pointing this out. First, we would like to correct our previous response: the binding ability of ALYREF to m^5^C-modified RNA was initially reported in doi: 10.1038/cr.2017.55, (and not in doi: 10.1038/s41556-019-0361-y), where it was observed through PAR-CLIP analysis that the K171 mutation weakens its binding affinity to m^5^C -modified RNA.

Our previous experimental approach was not optimal: the protein concentration in the INPUT group was too high, leading to overexposure in the experimental group. Additionally, we did not conduct a quantitative analysis of the results at that time. In response to your suggestion, we performed RNA pull-down experiments with YBX1 and ALYREF, rather than with the pan-DRAM protein, to better validate and reproduce the previously reported findings. Our quantitative analysis revealed that both ALYREF and YBX1 exhibit a stronger affinity for m^5^C -modified RNAs. Furthermore, mutating the key amino acids involved in m^5^C recognition significantly reduced the binding affinity of both readers. These results align with previous studies (doi: 10.1038/cr.2017.55 and doi: 10.1038/s41556-019-0361-y), confirming that ALYREF and YBX1 are specific readers of m^5^C -modified RNAs. However, our detection system has certain limitations. Despite mutating the critical amino acids, both readers retained a weak binding affinity for m^5^C, suggesting that while the mutation helps reduce false positives, it is still challenging to precisely map the distribution of m^5^C modifications. To address this, we plan to further investigate the protein structure and function to obtain a more accurate m^5^C sequencing of the transcriptome in future studies. Accordingly, we have updated our results and conclusions in lines 294-299 and discuss these limitations in lines 109114.

In addition, while the m^5^C assay can be performed using only the DRAM system alone, comparing it with the DRAM^mut^ control enhances the accuracy of m^5^C region detection. To minimize the variations in transfection efficiency across experimental groups, it is recommended to use the same batch of transfections. This approach not only ensures more consistent results but also improve the standardization of the DRAM assay, as discussed in the section added on line 308-312.

(2) Using sodium arsenite treatment of cells as a means to change the m^5^C status of transcripts through the downregulation of the two major m^5^C writer proteins NSUN2 and NSUN6 is problematic and the conclusions from these experiments are not warranted. Sodium arsenite is a chemical that poisons every protein containing thiol groups. Not only do NSUN proteins contain cysteines but also the base editor fusion proteins. Arsenite will inactivate these proteins, hence the editing frequency will drop, as observed in the experiments shown in Figure 5, which the authors explain with fewer m^5^C sites to be detected by the fusion proteins.The authors have not addressed the point made by this reviewer. Instead the authors state that they have not addressed that possibility. They claim that they have revised the results section, but this reviewer can only see the point raised in the conclusions. An experiment would have been to purify base editors via the HA tag and then perform some kind of binding/editing assay in vitro before and after arsenite treatment of cells.

We appreciate the reviewer’s insightful comment. We fully agree with the concern raised. In the original manuscript, our intention was to use sodium arsenite treatment to downregulate NSUN mediated m^5^C levels and subsequently decrease DRAM editing efficiency, with the aim of monitoring m^5^C dynamics through the DRAM system. However, as the reviewer pointed out, sodium arsenite may inactivate both NSUN proteins and the base editor fusion proteins, and any such inactivation would likely result in a reduced DRAM editing.

This confounds the interpretation of our experimental data.

As demonstrated in Author response image 1A, western blot analysis confirmed that sodium arsenite indeed decreased the expression of fusion proteins. In addition, we attempted in vitro fusion protein purificationusing multiple fusion tags (HIS, GST, HA, MBP) for DRAM fusion protein expression, but unfortunately, we were unable to obtain purified proteins. However, using the Promega TNT T7 Rapid Coupled In Vitro Transcription/Translation Kit, we successfully purified the DRAM protein (Author response image 1B). Despite this success, subsequent in vitro deamination experiments did not yield the expected mutation results (Author response image 1C), indicating that further optimization is required. This issue is further discussed in line 314-315.

Taken together, the above evidence supports that the experiment of sodium arsenite treatment was confusing and we determined to remove the corresponding results from the main text of the revised manuscript.

**Author response image 1. sa2fig1:** 

(3) The authors should move high-confidence editing site data contained in Supplementary Tables 2 and 3 into one of the main Figures to substantiate what is discussed in Figure 4A. However, the data needs to be visualized in another way then excel format. Furthermore, Supplementary Table 2 does not contain a description of the columns, while Supplementary Table 3 contains a single row with letters and numbers.The authors have not addressed the point made by this reviewer. Figure 3F shows the screening process for DRAM-seq assays and principles for screening highconfidence genes rather than the data contained in Supplementary Tables 2 and 3 of the former version of this manuscript.

Thank you for your valuable suggestion. We have visualized the data from Supplementary Tables 2 and 3 in Figure 4A as a circlize diagram (described in lines 213-216), illustrating the distribution of mutation sites detected by the DRAM system across each chromosome. Additionally, to improve the presentation and clarity of the data, we have revised Supplementary Tables 2 and 3 by adding column descriptions, merging the DRAM-ABE and DRAM-CBE sites, and including overlapping m^5^C genes from previous datasets.

**Responses to Reviewer’s Comments:**

**To Reviewer #3:**
The authors have again tried to address the former concern by this reviewer who questioned the specificity of both m^5^C reader proteins towards modified RNA rather than unmodified RNA. The authors chose to do RNA pull down experiments which serve as a proxy for proving the specificity of ALYREF and YBX1 for m^5^C modified RNAs. Even though this reviewer asked for determining the enrichment factor of the reader-base editor fusion proteins (as wildtype or mutant for the identified m^5^C specificity motif) when presented with m^5^C-modified RNAs, the authors chose to use both reader proteins alone (without the fusion to an editor) as wildtype and as respective m^5^C-binding mutant in RNA in vitro pull-down experiments along with unmodified and m^5^C-modified RNA oligomers as binding substrates. The quantification of these pull-down experiments (n=2) have now been added, and are revealing that (according to SFigure 1 E and G) YBX1 enriches an RNA containing a single m^5^C by a factor of 1.3 over its unmodified counterpart, while ALYREF enriches by a factor of 4x. This is an acceptable approach for educated readers to question the specificity of the reader proteins, even though the quantification should be performed differently (see below).Given that there is no specific sequence motif embedding those cytosines identified in the vicinity of the DRAM-edits (Figure 3J and K), even though it has been accepted by now that most of the m^5^C sites in mRNA are mediated by NSUN2 and NSUN6 proteins, which target tRNA like substrate structures with a particular sequence enrichment, one can conclude that DRAM-Seq is uncovering a huge number of false positives. This must be so not only because of the RNA bisulfite seq data that have been extensively studied by others, but also by the following calculations: Given that the m^5^C/C ratio in human mRNA is 0.02-0.09% (measured by mass spec) and assuming that 1/4 of the nucleotides in an average mRNA are cytosines, an mRNA of 1.000 nucleotides would contain 250 Cs. 0.02- 0.09% m^5^C/C would then translate into 0.05-0.225 methylated cytosines per 250 Cs in a 1000 nt mRNA. YBX1 would bind every C in such an mRNA since there is no m^5^C to be expected, which it could bind with 1.3 higher affinity. Even if the mRNAs would be 10.000 nt long, YBX1 would bind to half a methylated cytosine or 2.25 methylated cytosines with 1.3x higher affinity than to all the remaining cytosines (2499.5 to 2497.75 of 2.500 cytosines in 10.000 nt, respectively). These numbers indicate a 4999x to 1110x excess of cytosine over m^5^C in any substrate RNA, which the "reader" can bind as shown in the RNA pull-downs on unmodified RNAs. This reviewer spares the reader of this review the calculations for ALYREF specificity, which is slightly higher than YBX1. Hence, it is up to the capable reader of these calculations to follow the claim that this minor affinity difference allows the unambiguous detection of the few m^5^C sites in mRNA be it in the endogenous scenario of a cell or as fusion-protein with a base editor attached?

We sincerely appreciate the reviewer’s rigorous analysis. We would like to clarify that in our RNA pulldown assays, we indeed utilized the full DRAM system (reader protein fused to the base editor) to reflect the specificity of m^5^C recognition. As previously suggested by the reviewer, to independently validate the m^5^C-binding specificity of ALYREF and YBX1, we performed separate pulldown experiments with wild-type and mutant reader proteins (without the base editor fusion) using both unmodified and m^5^C-modified RNA substrates. This approach aligns with established methodologies in the field (doi:10.1038/cr.2017.55 and doi: 10.1038/s41556-019-0361-y). We have revised the Methods section (line 230) to explicitly describe this experimental design.

Although the m^5^C/C ratios in LC/MS-assayed mRNA are relatively low (ranging from 0.02% to 0.09%), as noted by the reviewer, both our data and previous studies have demonstrated that ALYREF and YBX1 preferentially bind to m^5^C-modified RNAs over unmodified RNAs, exhibiting 4-fold and 1.3-fold enrichment, respectively (Supplementary Figure 1E–1G). Importantly, this specificity is further enhanced in the DRAM system through two key mechanisms: first, the fusion of reader proteins to the deaminase restricts editing to regions near m^5^C sites, thereby minimizing off-target effects; second, background editing observed in reader-mutant or deaminase controls (e.g., DRAM^mut^-CBE in Figure 2D) is systematically corrected for during data analysis.

We agree that the theoretical challenge posed by the vast excess of unmodified cytosines. However, our approach includes stringent controls to alleviate this issue. Specifically, sites identified in NSUN2/NSUN6 knockout cells or reader-mutant controls are excluded (Figure 3F), which significantly reduces the number of false-positive detections. Additionally, we have observed deamination changes near high-confidence m^5^C methylation sites detected by RNA bisulfite sequencing, both in first-generation and high-throughput sequencing data. This observation further substantiates the validity of DRAM-Seq in accurately identifying m^5^C sites.

We fully acknowledge that residual false positives may persist due to the inherent limitations of reader protein specificity, as discussed in line 299-301 of our manuscript. To address this, we plan to optimize reader domains with enhanced m^5^C binding (e.g., through structure-guided engineering), which is also previously implemented in the discussion of the manuscript.

The reviewer supports the attempt to visualize the data. However, the usefulness of this Figure addition as a readable presentation of the data included in the supplement is up to debate.

Thank you for your kind suggestion. We understand the reviewer's concern regarding data visualization. However, due to the large volume of DRAM-seq data, it is challenging to present each mutation site and its characteristics clearly in a single figure. Therefore, we chose to categorize the data by chromosome, which not only allows for a more organized presentation of the DRAM-seq data but also facilitates comparison with other database entries. Additionally, we have updated Supplementary Tables 2 and 3 to provide comprehensive information on the mutation sites. We hope that both the reviewer and editors will understand this approach. We will, of course, continue to carefully consider the reviewer's suggestions and explore better ways to present these results in the future.

(3) A set of private Recommendations for the Authors that outline how you think the science and its presentation could be strengthened

NEW COMMENTS to TEXT:Abstract:"5-Methylcytosine (m^5^C) is one of the major post-transcriptional modifications in mRNA and is highly involved in the pathogenesis of various diseases."In light of the increasing use of AI-based writing, and the proof that neither DeepSeek nor ChatGPT write truthfully statements if they collect metadata from scientific abstracts, this sentence is utterly misleading.m^5^C is not one of the major post-transcriptional modifications in mRNA as it is only present with a m^5^C/C ratio of 0.02- 0.09% as measured by mass-spec. Also, if m^5^C is involved in the pathogenesis of various diseases, it is not through mRNA but tRNA. No single published work has shown that a single m^5^C on an mRNA has anything to do with disease. Every conclusion that is perpetuated by copying the false statements given in the many reviews on the subject is based on knock-out phenotypes of the involved writer proteins. This reviewer wishes that the authors would abstain from the common practice that is currently flooding any scientific field through relentless repetitions in the increasing volume of literature which perpetuate alternative facts.

We sincerely appreciate the reviewer’s insightful comments. While we acknowledge that m^5^C is not the most abundant post-transcriptional modification in mRNA, we believe that research into m^5^C modification holds considerable value. Numerous studies have highlighted its role in regulating gene expression and its potential contribution to disease progression. For example, recent publications have demonstrated that m^5^C modifications in mRNA can influence cancer progression, lipid metabolism, and other pathological processes (e.g., PMID: 37845385; 39013911; 39924557; 38042059; 37870216).

We fully agree with the reviewer on the importance of maintaining scientific rigor in academic writing. While m^5^C is not the most abundant RNA modification, we cannot simply draw a conclusion that the level of modification should be the sole criterion for assessing its biological significance. However, to avoid potential confusion, we have removed the word “major”.

COMMENTS ON FIGURE PRESENTATION:Figure 2D:The main text states: "DRAM-CBE induced C to U editing in the vicinity of the m^5^C site in AP5Z1 mRNA, with 13.6% C-to-U editing, while this effect was significantly reduced with APOBEC1 or DRAM^mut^-CBE (Fig.2D)." The Figure does not fit this statement. The seq trace shows a U signal of about 1/3 of that of C (about 30%), while the quantification shows 20+ percent

Thank you for your kind suggestion. Upon visual evaluation, the sequencing trace in the figure appears to suggest a mutation rate closer to 30% rather than 22%. However, relying solely on the visual interpretation of sequencing peaks is not a rigorous approach. The trace on the left represents the visualization of Sanger sequencing results using SnapGene, while the quantification on the right is derived from EditR 1.0.10 software analysis of three independent biological replicates. The C-to-U mutation rates calculated were 22.91667%, 23.23232%, and 21.05263%, respectively. To further validate this, we have included the original EditR analysis of the Sanger sequencing results for the DRAM-CBE group used in the left panel of Figure 2D (see Author response image 2). This analysis confirms an m^5^C fraction (%) of 22/(22+74) = 22.91667, and the sequencing trace aligns well with the mutation rate we reported in Figure 2D. In conclusion, the data and conclusions presented in Figure 2D are consistent and supported by the quantitative analysis.

Figure 4B: shows now different numbers in Venn-diagrams than in the same depiction, formerly Figure 4A

We sincerely thank the reviewer for pointing out this issue, and we apologize for not clearly indicating the changes in the previous version of the manuscript. In response to the initial round of reviewer comments, we implemented a more stringent data filtering process (as described in Figure 3F and method section) : "For high-confidence filtering, we further adjusted the parameters of Find_edit_site.pl to include an edit ratio of 10%–60%, a requirement that the edit ratio in control samples be at least 2-fold higher than in NSUN2 or NSUN6knockout samples, and at least 4 editing events at a given site." As a result, we made minor adjustments to the Venn diagram data in Figure 4A, reducing the total number of DRAM-edited mRNAs from 11,977 to 10,835. These changes were consistently applied throughout the manuscript, and the modifications have been highlighted for clarity. Importantly, these adjustments do not affect any of the conclusions presented in the manuscript.

Figure 4B and D: while the overlap of the DRAM-Seq data with RNA bisulfite data might be 80% or 92%, it is obvious that the remaining data DRAM seq suggests a detection of additional sites of around 97% or 81.83%. It would be advised to mention this large number of additional sites as potential false positives, unless these data were normalized to the sites that can be allocated to NSUN2 and NSUN6 activity (NSUN mutant data sets could be substracted).

Thank you for pointing this out. The Venn diagrams presented in Figure 4B and D already reflect the exclusion of potential false-positive sites identified in methyltransferasedeficient datasets, as described in our experimental filtering process, and they represent the remaining sites after this stringent filtering. However, we acknowledge that YBX1 and ALYREF, while preferentially binding to m^5^C-modified RNA, also exhibit some affinity for unmodified RNA. Although we employed rigorous controls, including DRAM^mut^ and deaminase groups, to minimize false positives, the possibility of residual false positives cannot be entirely ruled out. Addressing this limitation would require even more stringent filtering methods, as discussed in lines 299–301 of the manuscript. We are committed to further optimizing the DRAM system to enhance the accuracy of transcriptome-wide m^5^C analysis in future studies.

SFigure 1: It is clear that the wild type version of both reader proteins are robustly binding to RNA that does not contain m^5^C. As for the calculations of x-fold affinity loss of RNA binding using both ALYREF -mut or YBX1 -mut, this reviewer asks the authors to determine how much less the mutated versions of the proteins bind to a m^5^C-modified RNAs. Hence, a comparison of YBX1 versus YBX1 -mut (ALYREF versus ALYREF -mut) on the same substrate RNA with the same m^5^C-modified position would allow determining the contribution of the so-called modification binding pocket in the respective proteins to their RNA binding. The way the authors chose to show the data presently is misleading because what is compared is the binding of either the wild type or the mutant protein to different RNAs.

We appreciate the reviewer’s valuable feedback and apologize for any confusion caused by the presentation of our data. We would like to clarify the rationale behind our approach. The decision to present the wild-type and mutant reader proteins in separate panels, rather than together, was made in response to comments from Reviewer 2. Below, we provide a detailed explanation of our experimental design and its justification.

First, we confirmed that YBX1 and ALYREF exhibit stronger binding affinity to m^5^Cmodified RNA compared to unmodified RNA, establishing their role as m^5^C reader proteins. Next, to validate the functional significance of the DRAM^mut^ group, we demonstrated that mutating key amino acids in the m^5^C-binding pocket significantly reduces the binding affinity of YBX1^mut^ and ALYREF^mut^ to m^5^C-modified RNA. This confirms that the DRAM^mut^ group effectively minimizes false-positive results by disrupting specific m^5^C interactions.

Crucially, in our pull-down experiments, both the wild-type and mutant proteins (YBX1/YBX1^mut^ and ALYREF/ALYREF^mut^) were incubated with the same RNA sequences. To avoid any ambiguity, we have included the specific RNA sequence information in the Methods section (lines 463–468). This ensures a assessment of the reduced binding affinity of the mutant versions relative to the wild-type proteins, even though they are presented in separate panels.

We hope this explanation clarifies our approach and demonstrates the robustness of our findings. We sincerely appreciate the reviewer’s understanding and hope this addresses their concerns.

SFigure 2C: first two panels are duplicates of the same image.

Thank you for pointing this out. We sincerely apologize for incorrectly duplicating the images. We have now updated Supplementary Figure 2C with the correct panels and have provided the original flow cytometry data for the first two images. It is important to note that, as demonstrated by the original data analysis, the EGFP-positive quantification values (59.78% and 59.74%) remain accurate. Therefore, this correction does not affect the conclusions of our study. Thank you again for bringing this to our attention.

**Author response image 3. sa2fig3:** 

SFigure 4B: how would the PCR product for NSUN6 be indicative of a mutation? The used primers seem to amplify the wildtype sequence.

Thank you for your kind suggestion. In our NSUN6^-/-^ cell line, the NSUN6 gene is only missing a single base pair (1bp) compared to the wildtype, which results in frame shift mutation and reduction in NSUN6 protein expression. We fully agree with the reviewer that the current PCR gel electrophoresis does not provide a clear distinction of this 1bp mutation. To better illustrate our experimental design, we have included a schematic representation of the knockout sequence in SFigure 4B. Additionally, we have provided the original sequencing data, and the corresponding details have been added to lines 151-153 of the manuscript for further clarification.

**Author response image 4. sa2fig4:** 

SFigure 4C: the Figure legend is insufficient to understand the subfigure.

Thank you for your valuable suggestion. To improve clarity, we have revised the figure legend for SFigure 4C, as well as the corresponding text in lines 178-179. We have additionally updated the title of SFigure 4 for better clarity. The updated SFigure 4C now demonstrates that the DRAM-edited mRNAs exhibit a high degree of overlap across the three biological replicates.

SFigure 4D: the Figure legend is insufficient to understand the subfigure.

Thank you for your kind suggestion. We have revised the figure legend to provide a clearer explanation of the subfigure. Specifically, this figure illustrates the motif analysis derived from sequences spanning 10 nucleotides upstream and downstream of DRAMedited sites mediated by loci associated with NSUN2 or NSUN6. To enhance clarity, we have also rephrased the relevant results section (lines 169-175) and the corresponding discussion (lines 304-307).

SFigure 7: There is something off with all 6 panels. This reviewer can find data points in each panel that do not show up on the other two panels even though this is a pairwise comparison of three data sets (file was sent to the Editor) Available at https://elife-rp.msubmit.net/elife-rp_files/2025/01/22/00130809/02/130809_2_attach_27_15153.pdf

Response: We thank the reviewer for pointing this out. We would like to clarify the methodology behind this analysis. In this study, we conducted pairwise comparisons of the number of DRAM-edited sites per gene across three biological replicates of DRAM-ABE or DRAM-CBE, visualized as scatterplots. Each data point in the plots corresponds to a gene, and while the same gene is represented in all three panels, its position may vary vertically or horizontally across the panels. This variation arises because the number of mutation sites typically differs between replicates, making it unlikely for a data point to occupy the exact same position in all panels. A similar analytical approach has been used in previous studies on m6A (PMID: 31548708). To address the reviewer’s concern, we have annotated the corresponding positions of the questioned data points with arrows in Author response image 5.

**Author response image 5. sa2fig5:**